# Magnitude Distance: A Geometric Measure of Dataset Similarity

**Sahel Torkamani** [1]  **Henry Gouk** [1]  **Rik Sarkar** [1]

## Abstract

Quantifying the distance between datasets is a fundamental question in mathematics and machine learning. We propose *magnitude distance*, a novel distance metric defined on finite datasets using the notion of the *magnitude* of a metric space. The proposed distance incorporates a tunable scaling parameter, $t$, that controls the sensitivity to global structure (small $t$) and finer details (large $t$). We prove several theoretical properties of magnitude distance, including its limiting behavior across scales and conditions under which it satisfies key metric properties. In contrast to classical distances, we show that magnitude distance remains discriminative in high-dimensional settings when the scale is appropriately tuned. We further demonstrate how magnitude distance can be used as a training objective for push-forward generative models. Our experimental results support our theoretical analysis and demonstrate that magnitude distance provides meaningful signals, comparable to established distance-based generative approaches.

## 1. Introduction

This paper introduces the *magnitude distance*, a novel distance measuring the dissimilarity between finite sets $X, Y \in \mathbb{R}^D$, which captures geometric properties of the dataset. The need for quantitatively comparing datasets arises in a broad swathe of machine learning and data analysis applications ranging from hypothesis testing (Gretton et al., 2012) to training generative models (Arjovsky et al., 2017). However, all choices of distance measure between probability distributions, or finite samples from such distributions, make some tradeoff between a large number of desirable properties. One must decide whether robustness

to outliers can be compromised in order for increased sensitivity to differences between distributions, whether the underlying geometric structure of the space can be ignored in order to improve the estimator's sample efficiency, and many other potential compromises.

The distance is built on the magnitude of metric spaces introduced by Leinster (2008). Magnitude is a relatively new isometric invariant of metric spaces. Intuitively, it can be seen as measuring the "effective size" of mathematical objects. It has been defined, adapted, and studied in many different contexts such as topology, finite metric spaces, entropy, compact metric spaces, graphs, and machine learning (Leinster, 2013; 2021; Barceló & Carbery, 2018; Kaneta & Yoshinaga, 2021; Giusti & Menara, 2024). Magnitude distance inherits properties of metric magnitude and is able to leverage the topological and geometric properties of datasets. A distinguishing property of the magnitude distance is that it has a scaling parameter, $t$, that controls the sensitivity to capturing local differences or global structures in data. It satisfies several other properties that are desirable for a distance between point sets. In high dimensions, where classical distances such as Wasserstein suffer from severe concentration and loss of discriminability, magnitude distance remains sensitive to structural differences when $t$ is appropriately tuned, due to its kernel-inverted spectral behavior that emphasizes local geometry over global collapse.

To demonstrate the utility of the proposed approach, we develop a proof of concept push-forward generative model inspired by the principles of curriculum learning (CL), which we refer to as a Magnitude Generative Network (MagGN). CL, motivated by the idea of a curriculum in human learning, imposes structure on the training set in order to expose the model to "easier" examples before "harder" ones. Empirically, CL has been shown to accelerate and improve the learning process in many machine learning paradigms (Selfridge et al., 1985; Bengio et al., 2009). Traditional CL approaches require an initial sorting of training examples by difficulty. In contrast, MagGN tunes the complexity of examples by scaling them within the space. Intuitively, by starting with a small scaling parameter, we enable the model to learn coarse structures first. As we gradually increase the scale, points become spread apart, allowing the progressive capture of finer-grained details.

---

[1]School of Informatics, University of Edinburgh, Location, Country. Correspondence to: Sahel Torkamani <sahel.torkamani@ed.ac.uk>, Henry Gouk <henry.gouk@ed.ac.uk>, Rik Sarkar <rsarkar@inf.ed.ac.uk>.

*Proceedings of the $43^{rd}$ International Conference on Machine Learning*, Seoul, South Korea. PMLR 306, 2026.

We summarize our contributions in the following:

- In Section 4, we define the magnitude distance $d^t_{Mag}$ and its normalized variant for finite sets in $\mathbb{R}^D$. In the process of doing this, we extend existing results on the magnitude of metric spaces from non-redundant sets to all sets in Theorem 4.1.

- In Section 5, we analyze various properties of $d^t_{Mag}$, beginning with the metric axioms in Theorem 5.2. We prove non-negativity, and conditional identity of indiscernibles under a proposed notion of *magnitude equivalence*, and demonstrate the lack of a triangle inequality in general. We also study the behaviour of $d^t_{Mag}$ over different values of the scaling parameter in Theorem 5.3. We demonstrate both theoretically and practically that $d^t_{Mag}$ remains sensitive to local differences when the scaling parameter $t$ is appropriately tuned. In Theorem 5.5 we prove the global boundedness for outlier robustness.

- In Section 6, we show that magnitude distance can be used as a learning signal for generative modelling by proposing MagGN, a push-forward generative modelling technique that uses multi-scale magnitude distance as the loss function.

## 2. Related Work

### 2.1. Magnitude of Metric Spaces

Magnitude is broad concept relevant to various areas in mathematics. Leinster (2008) defined it in a categorical form as an Euler characteristic. Since then, it has been studied in the context of topology (Kaneta & Yoshinaga, 2021), graphs (Leinster, 2019; Giusti & Menara, 2024), and various other topics in mathematics. In machine learning, the geometric properties of magnitude have recently been applied in various ways, such as for clustering (O'Mally, 2023) and diversity measurement (Limbeck et al., 2024). In Andreeva et al. (2023; 2024), magnitude dimension is used to study generalization in neural networks. Efficient approximation methods have been introduced by Andreeva et al. (2025).

### 2.2. Dataset and Distribution Distances

A number of distribution distances have meaningful interpretations when defined over finite sets of data, all of which make different tradeoffs related to scalability with the data dimension, sample efficiency, outlier robustness, and many other factors. Two measures that are particularly related to our proposed Magnitude distance include the Maximum Mean Discrepancy (MMD) (Gretton et al., 2012) and the Wasserstein distance (Givens & Shortt, 1984). Both of these distances can leverage the underlying geometric structure of the metric space in which the data reside. MMD accomplishes this through the use of an appropriately chosen kernel function, while Wasserstein distance relies on the ambient metric of the space. We elucidate in Section 5.3 the connection between our proposed approach and MMD with the exponential kernel in terms of the spectra of the corresponding kernel matrix. We also explore the outlier robustness of our approach and discuss how this compares with the Wasserstein distance.

### 2.3. Push-Forward Generative Models

To demonstrate the utility of our proposed distance measure, we show how it can be used as the training objective of a push-forward generative modelling approach (Salmona et al., 2022). Push-forward generative modelling is a broad group of approaches that generate data points by first sampling from a tractable reference distribution (e.g., multivariate normal or uniform) and then passing this sample through a learned map. Many algorithms that are used for learning this map can be interpreted as finding a learned map that minimizes an objective based on a dataset distance metric. Generative Adversarial Networks (Goodfellow et al., 2014) attempt to find a distribution that is close in terms of Jensen-Shannon divergence, and extensions such as Wasserstein GANs (Arjovsky et al., 2017) generalize this to metrics like the Wasserstein Distance. Perhaps most related to our case study are the set of approaches that leverage MMD as the objective; this includes the MMD GAN (Li et al., 2017), which minimizes the MMD between generated data and real data, and the Inductive Moment Matching approach of Zhou et al. (2025) that uses MMD to learn a score function from which the push-forward map is constructed.

## 3. Preliminaries

For a finite metric space, $(X, d)$, we define the *similarity matrix* as $\zeta_X(x_i, x_j) := \exp(-d(x_i, x_j))$, for every $x_i, x_j \in X$. The concept of *magnitude*, defined in terms of a weighting, is given below.

**Definition 3.1** (Metric Magnitude). A weighting of $(X, d)$ is a function $w_X : X \rightarrow \mathbb{R}$ satisfying $\sum_{j \in X} \zeta_X(x_i, x_j)\, w_X(x_j) = 1$ for every $x_i \in X$, where $w_X(x_i)$ is called the *magnitude weight*. The magnitude of $(X, d)$ is defined as

$$Mag(X, d) := \sum_{x_i \in X} w_X(x_i). \tag{1}$$

In general, the existence of a suitable weighting, and therefore the magnitude, is not guaranteed. However, if $X$ is a finite subset of $\mathbb{R}^D$, then $\zeta_X$ is a symmetric positive definite matrix [Theorem 2.5.3](Leinster, 2013). In particular,

$\zeta_X$ is invertible so a unique weighting exists and the magnitude is well defined. In this case the weighting vector can be computed by inverting the similarity matrix $\zeta_X$ and summing all the entries; i.e., $w_X := \zeta_X^{-1} \mathbb{1}$, where $\mathbb{1}$ is the $|X| \times 1$ column vector of all ones. Consequently, magnitude is the sum of all the entries of the weighting vector, i.e., $Mag(X) := \mathbb{1}^* w_X = \mathbb{1}^* \zeta_X^{-1} \mathbb{1}$. When the distance measure, $d$, is understood from context, such as in $\mathbb{R}^D$, we often omit it in the notation.

One can introduce a parameter, $t \in \mathbb{R}_+$, to define the *scaled metric space* $(tX, d_t)$, often denoted by $tX$. This is the metric with the same points as $X$ and metric $d_t(x, y) := t \cdot d(x, y)$. The *magnitude function* assigns each finite metric space $X$ to a family of scaled metric spaces $\{tX\}_{t>0}$.

**Definition 3.2** (Magnitude function). The *magnitude function* of a metric space $X$ is given by

$$\text{Mag}_X(t) = Mag(tX), \tag{2}$$

with the associated weighting vector denoted by $\mathbf{w}_X^t$.

## 4. Magnitude Distance

In the literature, a metric space is understood to be a *set* of distinct points—i.e., without duplicates. In the following, we extend the notion of magnitude to finite *collections* of points that may contain duplicates. In this case, while the weighting vector may not be unique, the magnitude remains well-defined.

**Theorem 4.1.** *Let $X$ be a finite collection of points in Euclidean space $\mathbb{R}^D$, and define the similarity matrix as before. Then:*

1. *If the elements of $X$ are distinct, then the similarity matrix is symmetric positive definite. Therefore, the inverse exists, and the weighting vector and magnitude are uniquely well-defined [Theorem 2.5.3] (Leinster, 2013).*

2. *If $X$ contains duplicate points, the similarity matrix is symmetric positive semidefinite (but not definite). However, the magnitude of $X$ is equal to that of $X'$, where $X'$ is the set of distinct elements in $X$. Also, while the weighting vector is not unique, all valid weightings for $X$ can be defined by distributing each $\mathbf{w}_{X'}(i)$ among the duplicates of $x_i$ (so that the coefficients sum to 1) where $\mathbf{w}_{X'}$ is the unique weighting on $X'$.*

Theorem 4.1 tells us magnitude is insensitive to redundancy. We therefore let $X, Y \subset \mathbb{R}^D$ be finite sets of points in Euclidean space, and the magnitude is computed disregarding sample redundancy within the sets. With this in place, we define the *magnitude distance*, which captures the dissimilarity between $X$ and $Y$ using magnitudes at a given scale, $t$.

**Definition 4.2** (Magnitude Distance). For two finite sets $X, Y \subset \mathbb{R}^D$, the magnitude distance with scale parameter $t \in \mathbb{R}_+$ is defined as

$$d_{Mag}^t(X, Y) = 2 \, \text{Mag}_{X \cup Y}(t) - \text{Mag}_X(t) - \text{Mag}_Y(t), \tag{3}$$

and the normalized magnitude distance is defined as $\tilde{d}_{Mag}^t(X, Y) = \frac{d_{Mag}^t(X,Y)}{\text{Mag}_{X \cup Y}(t)}$.

The proposed magnitude distance depends on the scaling parameter, $t$, that controls the sensitivity of $d_{Mag}^t$ to the separation between points. Intuitively, small $t$ focuses on the structure of the entire data, while large $t$ places more emphasis on the local differences and sample variability. Figure 1 illustrates the impact of the scaling parameter $t$ in a 100 dimensional space.

## 5. Properties

### 5.1. Metric Axioms

We now examine whether the magnitude distance satisfies the standard axioms of a metric. Before analyzing the axioms, we need to introduce the notion of *magnitude equivalence*.

**Definition 5.1** (Magnitude Equivalence). Let $X, Y \subset \mathbb{R}^D$ be finite sets with weighting vectors $\mathbf{w}_X^t$ and $\mathbf{w}_Y^t$ at scale $t$. The sets $X$ and $Y$ are *magnitude-equivalent* at scale $t$ if they have the same support of non-zero weight entries. In other words, we have $X \underset{\text{Mag}(t)}{=} Y$ if and only if

$$\{x \in X : \mathbf{w}_X^t(x) \neq 0\} = \{y \in Y : \mathbf{w}_Y^t(y) \neq 0\}. \tag{4}$$

Additional details on the properties of magnitude equivalence are provided in the Appendix B. Our main result regarding the metric properties of the magnitude distance is given below.

**Theorem 5.2.** *Magnitude distance $d_{Mag}^t$ has the following properties for $X, Y \subset \mathbb{R}^D$ and $t > 0$:*

- ***Symmetry:*** $d_{Mag}^t(X, Y) = d_{Mag}^t(Y, X)$ *by definition.*

- ***Non-negativity:*** *For any $t > 0$, we have $d_{Mag}^t(X, Y) \geq 0$.*

- ***Identity of indiscernibles:*** $d_{Mag}^t(X, Y) = 0 \iff X \underset{\text{Mag}(t)}{=} Y$.

- ***No Triangle inequality:*** $d_{Mag}^t$ *does not satisfy the triangle inequality in $\mathbb{R}^D$ for $D > 1$.*

### 5.2. Properties of Magnitude Distance Over values of $t$

As $t$ moves from 0 to $\infty$, we find that magnitude distance has a set of natural properties:

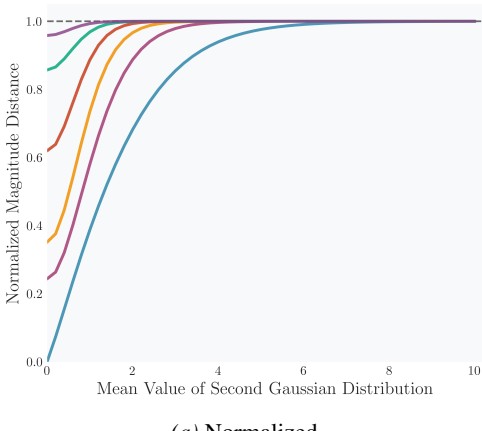
*(a)* Normalized

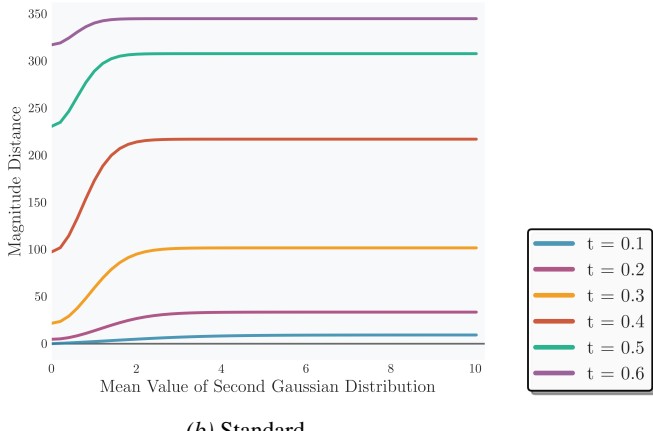
*(b)* Standard

*Figure 1.* Impact of the scaling parameter $t$, on the magnitude distance between samplings of $N(0,1)$ and $N(\mu, 1)$ in 100 dimensions. Each plot shows magnitude distance computed from 200 samples of $N(0,1)$ to 200 samples of $N(\mu, 1)$ with $t = 0.1$ (blue), 0.2 (purple), 0.3 (yellow), 0.4 (red), 0.5 (green), 0.6 (dark purple). Plot 1a and 1b show the normalized and standard magnitude distances, respectively. As the mean difference between samplings increases, both magnitude distances also increase. However, the standard magnitude distance converges toward different limits depending on $t$, while the normalized magnitude distance consistently converges to 1, regardless of $t$ By Theorem 5.3, these limits are also bounded above by the cardinality of the symmetric difference of the samples.

**Theorem 5.3.** *For every finite metric sets $X$ and $Y$, the magnitude distance $d^t_{Mag}(X, Y)$:*

- *Converges to 0 as $t \to 0$.*

- *Converges to the cardinality of $X \triangle Y$ as $t \to \infty$.*

- *For $t \gg 0$, the magnitude distance $d^t_{Mag}(X, Y)$ is increasing with respect to $t$.*

Similar properties have been shown for the magnitude function in [Proposition 2.2.6] (Leinster, 2013). Based on Theorem 5.3, in the following, we show how tuning $t$ controls the sensitivity of the distance to separation even in high dimensions.

### 5.3. Performance in High Dimensions

The magnitude distance operates through a kernelized form parameterized by a scale $t$, rather than relying on raw pairwise distances. Theorem 5.3 shows that as $t \to 0$, the magnitude distance converges to zero and as $t \to \infty$, the magnitude distance converges to $X \triangle Y$. Finite subsets of Euclidean space induce positive definite kernels, making the magnitude function $\text{Mag.}(t)$ well-defined and analytic over all $t > 0$. By the intermediate value theorem, as the scaling parameter $t$ varies over $(0, \infty)$, the magnitude distance attains every value between its limiting extremes.

**Proposition 5.4.** *For every two finite sets $X, Y \in \mathbb{R}^D$, and any value $\alpha \in (0, X \triangle Y)$, there exists a $t > 0$ such that $d^t_{Mag}(X, Y) = \alpha$, regardless of the ambient dimension.*

Thus, Proposition 5.4 indicates that high dimensionality alone does not force the distance to concentrate in a narrow

range. In the following, we show that an appropriate choice of $t$ prevents the distance from losing its discriminative power between distinct sets.

In high-dimensional settings, many distance functions suffer from loss of discriminability – where the pairwise distances collapse to a small range of values. In kernel-based distances, such as the Maximum Mean Discrepancy (MMD), this phenomenon can arise due to the spectral degeneracy of the kernel matrix. Given $n$ examples $x_1, \ldots, x_n \sim P(X)$ and $m$ examples from $y_1, \ldots, y_m \sim Q(Y)$, the empirical MMD statistic with kernel $k^t(x, y) = \exp(-td(x, y))$ (exponential kernel) is given by

$$\widehat{\text{MMD}}^2 = \frac{1}{n^2} \sum_{i,j \in [n]} \zeta_X(x_i, x_j) + \frac{1}{m^2} \sum_{i,j \in [m]} \zeta_Y(y_i, y_j)$$
$$- \frac{2}{nm} \sum_{i \in [n], j \in [m]} \zeta_{X \cup Y}(x_i, y_j),$$

where $\zeta_X$, $\zeta_Y$, and $\zeta_{X \cup Y}$ denote the kernel matrices (the similarity matrices defined in Section 3) induced by $k^t$ on the corresponding point sets. The magnitude distance, using the same kernel, is given by a related algebraic form, but depends on the inverse of the kernel matrix:

$$2 \sum_{i,j=1}^{n+m} \zeta_{X \cup Y}^{-1}(z_i, z_j) - \sum_{i,j=1}^{n} \zeta_X^{-1}(x_i, x_j) - \sum_{i,j=1}^{m} \zeta_Y^{-1}(y_i, y_j),$$

where $\zeta_X^{-1}$, $\zeta_Y^{-1}$, and $\zeta_{X \cup Y}^{-1}$ are the inverses of kernel matrices above and $z_i, z_j$ in $\zeta_{X \cup Y}^{-1}(z_i, z_j)$ can be any two points from $X \cup Y$. Magnitude distance and MMD differ fundamentally in how they aggregate spectral information. MMD computes a quadratic form based on the kernel matrix itself

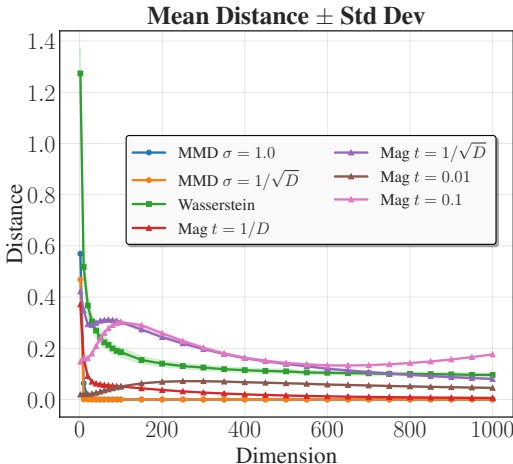

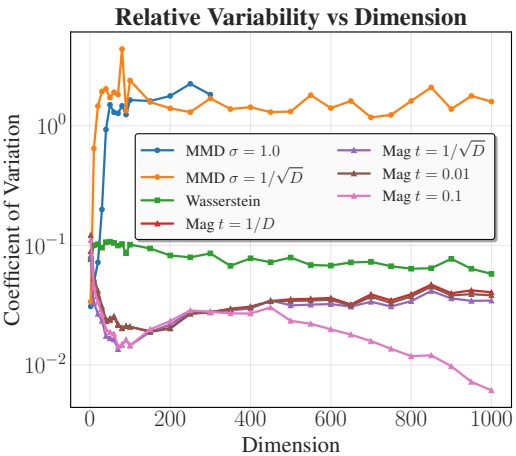

*Figure 2.* From two Gaussian distributions with identical covariance and a mean shift of 2, we compute the empirical MMD and magnitude distance between 500 samples from each distribution over 100 independent trials. The plot shows a comparison between the mean of empirical MMD distance with Gaussian kernel bandwidth $\sigma = 1$ and $\sigma = 1/\sqrt{D}$, and normalized magnitude distance for fixed kernel scales $t \in \{0.01, 0.1\}$ and adaptive scales $t = 1/D$ and $t = 1/\sqrt{D}$. MMD in both settings rapidly collapses toward zero as the dimension increases. Wasserstein distance decreases with dimension but remains comparatively stable. Magnitude distance with ($t = 1/D$) exhibits mis-scaling in low dimensions, producing overly small distances. For fixed scaling or scaling $t = 1/\sqrt{D}$, magnitude distance remains stable across dimensions, with only gradual changes.

$(\mathbf{1}^\top \zeta \mathbf{1} = \sum_i \lambda_i (\mathbf{1}^\top v_i)^2)$. In high dimensions, for commonly used characteristic kernels such as the Gaussian or exponential kernel, the kernel matrix tends to have low effective rank, and most eigenvalues collapse to near zero, except for a few dominant directions. Therefore, MMD becomes dominated by the top eigen-directions and projects the distance between the samples onto a low-dimensional subspace corresponding to these few top eigen-directions. In contrast, magnitude distance depends on the inverse kernel matrix, and the quadratic form $\mathbf{1}^\top \zeta^{-1} \mathbf{1} = \sum_i \lambda_i^{-1} (\mathbf{1}^\top v_i)^2$ is shaped by aggregating spectral information through inverse eigenvalue weighting. This prevents the distance from being dominated by the top eigen-directions and instead includes more directions corresponding to the remaining eigenvalues. Figure 2 shows that magnitude distance remains discriminative compared to empirical MMD distance in high dimensions. We note that the numerical scales of the normalized magnitude distance are not directly comparable with those of the Wasserstein distance or the MMD. Consequently, absolute values across different distances are not interpreted comparatively, and instead, we focus on the function's behavior as the dimension increases.

While the mean distance indicates the expected separation between the distributions, it does not capture the reliability

*Figure 3.* From two Gaussian distributions with identical covariance and a mean shift of 2, we compute empirical distances between 500 samples from each distribution over 100 independent trials. The plot shows the coefficient of variation (log-scale) of Wasserstein distance, MMD with Gaussian kernel bandwidths $\sigma = 1$ and $\sigma = 1/\sqrt{D}$, and normalized magnitude distance with fixed kernel scales $t \in \{0.01, 0.1\}$ and adaptive scales $t = 1/D$ and $t = 1/\sqrt{D}$. MMD shows unstable behavior across kernel choices, and Wasserstein distance has consistently higher relative variability than magnitude distance. Moreover, magnitude-distance adaptive scaling maintains lower relative variability across dimensions while avoiding collapse to 0. For fixed kernel scales, magnitude distance can initially show behavior similar to adaptive scaling; however, the outcome depends on the chosen scale. Larger fixed scale $t = 0.1$ enters an over-localized regime at sufficiently high dimensions, where off-diagonal kernel similarities vanish, and the distance collapses abruptly. Smaller fixed scale $t = 0.01$ preserves stability over a wider range of dimensions, although they are also expected to eventually fail as dimensionality continues to increase.

of this estimate. In high-dimensional spaces, empirical distances can exhibit substantial variability across samples, even when their expectations remain stable. To quantify this effect, we consider the coefficient of variation (CV), defined as the ratio of the standard deviation to the mean. Figure 3 reports the coefficient of variation of empirical distances as a function of dimension. Magnitude distance with adaptive scaling regimes ($t = D$, $t = \frac{1}{D}$), exhibits consistently lower relative variability than Wasserstein and MMD. Since the coefficient of variation is scale-invariant, it is unaffected by normalization or unit differences and therefore provides a fair comparison. In Figures 2 and 3, we report results using the sliced Wasserstein distance, a commonly used variation of the Wasserstein distance in high-dimensional settings (Deshpande et al., 2018).

## 5.4. Robustness to Outliers

In this section, we show magnitude distance is robust to outliers. Firstly, we prove that under natural assumptions, the distance is bounded.

**Theorem 5.5.** *Let $X, Y \subset \mathbb{R}^D$ be finite sets with nonnegative weighting vectors of $X, Y$, and $X \cup Y$. Then, we have:*

$$0 \leq d_{\text{Mag}}^t(X, Y) \leq 2(|X \cup Y|). \tag{5}$$

*where $|X|$ and $|Y|$ denote the number of points in $X$ and $Y$ respectively.*

Also, nonnegative weighting vectors are guaranteed in different finite metric spaces, such as all subsets of metric spaces when scaled up sufficiently, i.e., $t \gg 0$, and also $\mathbb{R}$, for which this global boundedness exists for any scaling parameter. A direct consequence of Theorem 5.5 is that the distance's sensitivity to adding or adjusting samples is also bounded. In contrast, most distances, such as the Wasserstein distance, are extremely sensitive to outliers, as their sensitivity to adding noise is not bounded, and a single outlier can arbitrarily increase the distance.

To demonstrate this outlier robustness, we generate two datasets, $B$ and $Y$, by sampling from normal distributions with different means. These are represented by blue points and yellow points, respectively. We also generate a third set of points, $Y'$, with much higher dispersion. These datasets are shown in Figure 4. We consider the magnitude distance and Wasserstein distance for two cases: the distance between $B$ and $Y$, and the distance between $B$ and $Y^* = Y \cup Y'$. The relative change in magnitude distance with $t = 20$ and $t = 5$ are $6.85\%$ and $10.29\%$ respectively, compared to $17.61\%$ for Wasserstein distance. Even for small values of $t$, the relative change in magnitude distance due to the addition of outliers remains smaller than that in the Wasserstein distance. This is because for points within some bounded region, sufficiently small values of $t$ cause all distances to be small. Moreover, this relative change decreases as $t$ increases.

To demonstrate robustness to outliers, we consider the classical Huber contamination model (Huber, 1992), in which most of the data comes from a "clean" distribution $P$ (e.g., Gaussian), while a small fraction $\epsilon$ is arbitrarily corrupted, coming from distribution $R$. The resulting data distribution is $Q = (1 - \epsilon)P + \epsilon R$ where a small fraction of samples is displaced to increasing distances while the majority of the distribution remains unchanged. As the outlier radius $R$ grows, the Wasserstein distance increases proportionally, reflecting the described sensitivity to extreme points. In contrast, Figure 5 shows that magnitude distance with appropriate tuning of the scaling parameter remains stably close to zero even as the outlier radius increases up to 1000. This robustness behavior of normalized magnitude distance is also illustrated in the Appendix E.

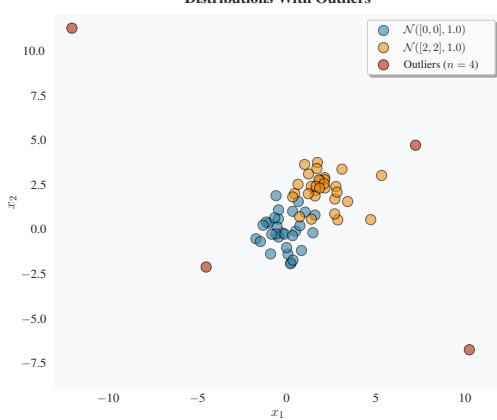

*Figure 4.* Outlier robustness in 2D with the baseline dataset $B \sim \mathcal{N}([0, 0], 1)$ (blue points), and set $Y \sim \mathcal{N}([2, 2], 1)$ (yellow points), with noisy variant $Y^*$, incorporating the outliers, $Y'$ (red points).

### 5.5. Computational Cost

The computational cost is determined mainly by the cost of computing $\text{Mag}_{X \cup Y}(t)$, which is dominated by the cost of matrix inversion. The best known bound for matrix inversion and multiplication is $\Omega(n^2 \log n)$ (Raz, 2002), but in practice, the method in Strassen (1969) (complexity $O(n^{2.81})$) and similar methods are often used. Faster practical approximations for magnitude computation were studied by Andreeva et al. (2025). It was shown that while magnitude is not submodular, greedy maximization based on Nemhauser et al. (1978) works well in practice. Further speedups can be achieved by using discrete center hierarchies to subsample the point set (Andreeva et al., 2025).

## 6. Application: Push-Forward Generative Modelling

As a demonstration of the utility of the magnitude distance, we introduce Magnitude Generative Network (MagGN), a novel push-forward generative model that incorporates the normalized magnitude distance $\hat{d}_{Mag}^t$ into the generator loss. Inspired by curriculum learning (CL), MagGN trains the model progressively, starting from an overall data structure and in a few targeted steps, gradually increasing the loss function's sensitivity to finer details. Technically, this is achieved by updating the loss function with greater values of $t$. For some $k \in \mathbb{N}$ and a list called t-steps $\{e_1, \ldots, e_k\}$, at epochs $e_i$ we add $\hat{d}_{Mag}^{t_i}(\mathcal{D}_r, \mathcal{D}_g)$ to the loss, respectively, with the magnitude scaling parameters $t_i$ where $t_1 \leq \cdots \leq t_k$.

$$l_e = \sum_{i : e_i \leq e} \hat{d}_{Mag}^{t_i}(\mathcal{D}_r, \mathcal{D}_g)$$

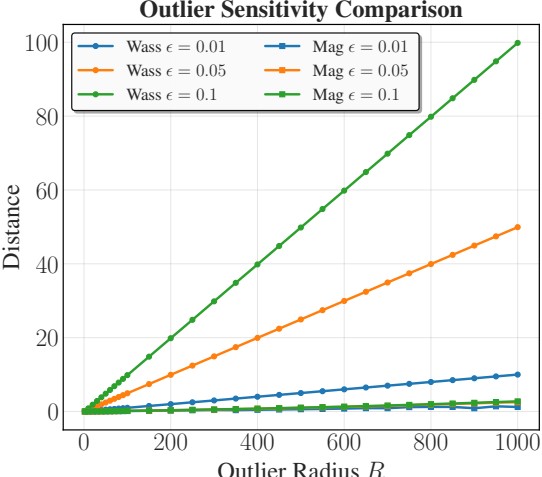

*Figure 5.* Outlier robustness under Huber contamination. We compare empirical Wasserstein distance and magnitude distance with $t = 0.001$ between 500 samples drawn from a contaminated distribution $Q = (1 - \epsilon)P + \epsilon R$ and 500 from the clean distribution $P$. The plot shows three contamination levels, $\epsilon \in \{0.01, 0.05, 0.1\}$. As the outlier radius increases, the Wasserstein distance grows, while magnitude distance remains close to 0.

where $\mathcal{D}_r$ and $\mathcal{D}_g$ denote the real and generated data, respectively. The normalized variant of the loss can be defined as $\bar{l}_e = l_e / |\{i : e_i \le e\}|$.

Although MagGN explicitly updates the loss function during training, this procedure can also be interpreted as progressively scaling the data via the parameters $t_i$. Increasing $t$ effectively increases the complexity of the data representation, aligning the training process with the principles of curriculum learning. Also, retaining the previous distances (smaller t values) ensures the model does not overfit to details.

## 7. Experiments

We conduct small scale experiments for the purpose of investigating two ideas: (i) determining whether the proof of concept push-forward generative model, MagGN, is a feasible direction for future research; and (ii) understanding the scaling behaviour of $t$, so we can provide some advice for how it should be tuned in practice. Our code is publicly available at https://github.com/saheltorkamani/Magnitude_Distance.

### 7.1. Performance of MagGN

We conduct experiments on the proposed MagGN model introduced in Section 6 to determine the feasibility of leveraging our novel metric in a generative modelling setting. WGAN and its variant with gradient penalty (WGAN-GP) (Gulrajani et al., 2017) are used as points of comparison,

since they are also push-forward generative models that do not require score-based simulation procedures. All the following experiments were performed on an NVIDIA L40S GPU, using CUDA 12.6.85 and PyTorch 2.5.1. All models are trained on the MNIST dataset. For fairness, we use the same multilayer perceptron (MLP) architecture and shared hyperparameters across all methods. These hyperparameters are tuned to optimize the performance of WGAN-GP. As a consequence, vanilla GAN, which typically has a different optimal configuration, underperforms in terms of sample quality. Therefore, we include both WGAN and WGAN-GP in our comparisons to separately account for computational efficiency and output quality. Also, all models were trained for the equivalent of 500 generator epochs. For WGAN and WGAN-GP, we use $n_{\text{critic}} = 5$, meaning that the critic (discriminator) is updated five times for each generator update. Figure 6 shows that across a range of $t$-values, MagGN achieves visual performance comparable to WGAN-GP and better than WGAN. Table 1 reports that MagGN reaches this performance more than 10 times faster than both WGAN and WGAN-GP. The Table also shows relative speedups when using vanilla WGAN as the baseline.

Limited texture and low visual complexity of MNIST make it a suitable dataset for MagGN, as it allows larger magnitude scales to be introduced without losing stability during training. Therefore, magnitude distance with these larger magnitude scales (such as 8 or 10) captures finer details, resulting in sharper generated images. To demonstrate the effectiveness of MagGN as a training objective on more complex datasets, we propose the necessary adjustments to enhance training on natural image datasets in Appendix F: Table 2 reports improved Inception Score (IS) and training time on CIFAR-10, where MagGN is used to pretrain the generator before fine-tuning with WGAN-GP, and Table 3 reports improved Fréchet Inception Distance (FID) on CelebA by implementation in VGG-16 feature space with a hybrid pixel–feature magnitude loss.

### 7.2. Tuning of Scaling Parameter $t$

While our results in Section 5.3 show that $d_{Mag}^t$ can remain discriminative in high-dimensional settings with a suitable scaling parameter $t$, selecting such a scale in practice is a nontrivial task. As observed, in Figures 2 and 3, fixed kernel scales grow with dimension and lead to mis-scaling in high dimensions. Adaptive cases, $t = 1/\sqrt{D}$ and $t = 1/D$, show a descent and stable behavior across dimensions; however, $(t = 1/D)$ can lead to mis-scaling in low dimensions. These observations suggest that $\mathcal{O}(1/\sqrt{D})$ provides a reasonable default scaling, consistent with our intuition, as it can compensate for the concentration of pairwise distances. However, a single scaling parameter is insufficient to capture all details, from local to global, in complex data. Motivated by this, we consider the aggregation of magnitude

*Table 1.* **Training time comparison on MNIST**

| Experiment | $t$ | $t$-steps | Training Time (s) | Speedup vs. WGAN (×) |
|---|---|---|---|---|
| WGAN | – | – | 66 064 | 1.00 |
| WGAN-GP | – | – | 77 520 | 0.85 |
| MagGN (normalized) | $\{0.01, 0.2, 2.0, 8.0\}$ | $\{1, 101, 251, 351\}$ | 5516 | 11.97 |
| MagGN (normalized) | $\{0.01, 0.3, 2.0, 8.0\}$ | $\{1, 101, 251, 351\}$ | 6029 | 10.95 |
| MagGN | $\{0.01, 0.3, 2.0, 10.0\}$ | $\{1, 101, 251, 351\}$ | 5440 | 12.14 |

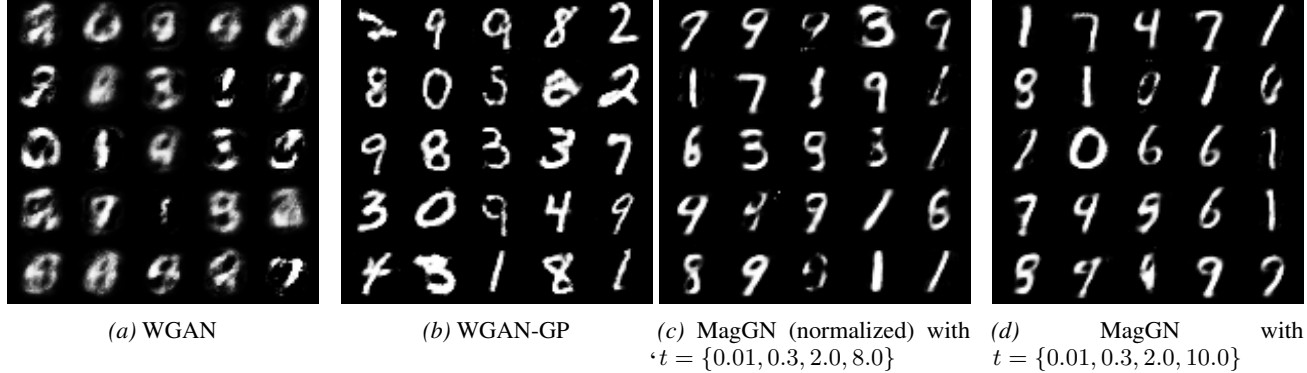

*(a)* WGAN     *(b)* WGAN-GP     *(c)* MagGN (normalized) with '$t = \{0.01, 0.3, 2.0, 8.0\}$     *(d)* MagGN with $t = \{0.01, 0.3, 2.0, 10.0\}$

*Figure 6.* MNIST samples generated after 500 generator epochs. (a) Vanilla WGAN. (b) WGAN-GP. (c) Normalized MagGN with $t = \{0.01, 0.3, 2.0, 8.0\}$ evaluated at $t$-steps $\{1, 101, 251, 351\}$. (d) MagGN with $t = \{0.01, 0.3, 2.0, 10.0\}$ evaluated at the same $t$-steps. The figure shows that MagGN generates digit samples that are visually competitive with the WGAN-GP baseline and sharper than vanilla WGAN.

distances computed on multiple scales, an approach that has also proven effective for MMD (Schrab et al., 2023) and is further demonstrated in Section 6.

Based on our empirical findings, we recommend selecting scaling parameters that grow geometrically. The aggregate scaling parameters approach effectively simplifies tuning, as the combination of smaller and larger values of $t$ simultaneously captures local and global data structures. In the MagGN experiments, we observed that while extremely large values of $t$ lead to mode collapse and over-localization, overly small values of $t$ also prevent the model from learning by providing insufficient structural information to the model. Importantly, if the scaling parameters are $\{t_1, \cdots t_k\}$ for some $k \in \mathbb{N}$, then recovery from an overly small initial scale $t_1$ is more difficult than from an overly large final scale $t_k$, since $t_1$ influences the loss function throughout the entire training process. Consequently, identifying an appropriate initial scale is the first and a crucial step. Based on the observations above, it is effective to initialize $t$ on the order of $1/\sqrt{D}$. For example, in the MNIST experiments, we used $t_1 = 0.01$, which is comparable to $1/\sqrt{D} = 1/28 \approx 0.03$. Moreover, the next scaling parameters can be selected using a geometric progression, followed by gradual refinement with smaller multiplicative increments. This strategy avoids overly large $t$, which could otherwise overwhelm the model when learning complex data structures. In the

MNIST experiments, this geometric growth corresponded to multiplicative factors $\{20, 10, 4\}$.

## 8. Conclusion

Difference between datasets is an important consideration in statistics and machine learning. In generative AI, it is of particular significance as a criterion in realistic data generation. Magnitude distance has several properties that are intuitive for dataset dissimilarity. In addition, it can be meaningfully computed in a resolution-dependent manner as required in a given circumstance. In this paper, we have demonstrated some of its essential properties, but further study will be needed for a broader understanding. Magnitude is closely related to various topological and geometric objects, as has been studied in mathematics, which can have a bearing on applications of magnitude distance. We expect that the magnitude distance defined here will find uses in various machine learning contexts as a fundamental tool. In particular, distances between datasets often find use-cases in areas such as differential privacy (Dwork et al., 2014), designing learning algorithms that provide robustness to distribution shifts (Ben-David et al., 2010), hypothesis testing (Gretton et al., 2012), and more.

## Impact Statement

This paper presents work whose goal is to advance the field of machine learning. There are many potential societal consequences of our work, none of which we feel must be specifically highlighted here.

## Acknowledgements

This project was supported by the Royal Academy of Engineering under the Research Fellowship programme.

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

## A. Proofs: Sets with Redundant Samples

*Proposition* (2.4.3, (Leinster, 2013)). Let $X$ be a positive definite finite metric space with similarity matrix $\zeta_X$. Then

$$Mag(X) \;=\; \sup_{v \neq 0} \frac{\left( \sum_{x \in X} v(x) \right)^2}{v^* \zeta_X v}, \tag{6}$$

where the supremum is taken over all nonzero vectors $v \in \mathbb{R}^X$, and $v^*$ denotes the transpose of $v$. Moreover, a vector $v$ attains the supremum if and only if it is a nonzero scalar multiple of the unique weighting on $X$.

**Theorem 4.1.** *Let $X$ be a finite collection of points in Euclidean space $\mathbb{R}^D$, and define the similarity matrix as before. Then:*

1. *If the elements of $X$ are distinct, then the similarity matrix is symmetric positive definite. Therefore, the inverse exists, and the weighting vector and magnitude are uniquely well-defined [Theorem 2.5.3] (Leinster, 2013).*

2. *If $X$ contains duplicate points, the similarity matrix is symmetric positive semidefinite (but not definite). However, the magnitude of $X$ is equal to that of $X'$, where $X'$ is the set of distinct elements in $X$. Also, while the weighting vector is not unique, all valid weightings for $X$ can be defined by distributing each $\mathbf{w}_{X'}(i)$ among the duplicates of $x_i$ (so that the coefficients sum to 1) where $\mathbf{w}_{X'}$ is the unique weighting on $X'$.*

*Proof of Theorem 4.1.* The first case follows directly from [Theorem 2.5.3] (Leinster, 2013), which shows that every finite subspace of Euclidean space is positive definite.

For the second case, let $X' = \{x_1, \cdots, x_n\}$ be the set of distinct elements in $X$, where each point $x_i$ appears $k_i$ times in $X$. Then, due to these duplications, the similarity matrix $\zeta_X$ is obtained from $\zeta_{X'}$ by repeating rows and columns corresponding to the multiplicities $k_i$. Therefore, $\zeta_X$ has linear dependencies between its rows (and columns) and $\text{rank}(\zeta_X) = \text{rank}(\zeta_{X'}) = n$. Thus, $\zeta_X$ is positive semidefinite and not definite.

Let $w_{X'}$ be the unique weighting on $X'$. We now show a valid weighting $w_X$ for $X$ by distributing each $\mathbf{w}_{X'}(j)$ arbitrarily among the $k_j$ duplicates of $x_j$. For each $x_j$, choose $\alpha_{j,1}, \cdots, \alpha_{j,k_j}$ such that $\sum_{l=1}^{k_j} \alpha_{j,l} = 1$. Then, define the weights of the duplicates $x_j^{(l)} \in X$ by

$$w_X(x_j^{(l)}) := \alpha_{j,l}\, w_{X'}(x_j), \qquad l = 1, \ldots, k_j$$

Now, we prove that $w_X$ is a valid weighting. For every $y \in X$ with representative $x_i \in X'$ we have:

$$\sum_{y' \in X} \zeta_X(y, y')\, w_X(y) = \sum_{x_j \in X'} \sum_{l=1}^{k_j} \zeta_{X'}(x_i, x_j)\, \alpha_{j,l} w_{X'}(x_j)$$

$$= \sum_{j \in X'} \zeta_{X'}(x_i, x_j) w_{X'}(x_j) \underbrace{\sum_{l=1}^{k_j} \alpha_{j,l}}_{=1}$$

$$= \sum_{j \in X'} \zeta_{X'}(x_i, x_j) w_{X'}(x_j) \underset{(a)}{=} 1.$$

where in $(a)$ we use the fact that $w_{X'}$ is a valid weighting for $X'$.

Moreover, we prove these are the only valid weightings for $X$ by assuming the opposite. That is, assume the weights of the duplicates $x_j^{(l)} \in X$ can be written as

$$w_X(x_j^{(l)}) := \alpha_{j,l}\, w_{X'}(x_j), \qquad l = 1, \ldots, k_j$$

where there exists $j$ such that $\sum_{l=1}^{k_j} \alpha_{j,l} \neq 1$. In this case, following the same approach, for every $y \in X$ with representative

$x_i \in X'$ we have:

$$1 = \sum_{y' \in X} \zeta_X(y, y') \, w_X(y) = \sum_{x_j \in X'} \sum_{l=1}^{k_j} \zeta_{X'}(x_i, x_j) \, \alpha_{j,l} w_{X'}(x_j)$$

$$= \sum_{j \in X'} \zeta_{X'}(x_i, x_j) \, \underbrace{w_{X'}(x_j) \sum_{l=1}^{k_j} \alpha_{j,l}}_{(b)}$$

then, for every $j$, the coefficient $w_{X'}(x_j) \sum_{l=1}^{k_j} \alpha_{j,l}$ should be a valid weighting for $X'$. But this contradicts the uniqueness of $w_{X'}$. Therefore, all valid weightings for $X$ can be defined by distributing each $\mathbf{w}_{X'}(i)$ among the duplicates of $x_i$, so that the coefficients sum to 1.

Finally, considering the magnitude defined as the sum of the entries of the weighting vector:

$$Mag(X) = \sum_{y \in X} w_X(y) = \sum_{x_j \in X'} \sum_{l=1}^{k_j} \alpha_{j,l} w_{X'}(x_j)$$

$$= \sum_{j \in X'} w_{X'}(x_j) \underbrace{\sum_{l=1}^{k_j} \alpha_{j,l}}_{=1}$$

$$= \sum_{j \in X'} w_{X'}(x_j) = Mag(X').$$

$\square$

**Corollary A.1** (Non-Negative Weighting). *Let $X$ be a finite collection of points in Euclidean space $\mathbb{R}^D$, and let $X'$ denote the set of distinct elements in $X$. If the unique weighting vector of $X'$ is non-negative, $X$ also has a non-negative weighting.*

## B. Proofs: Metric Axioms

**Lemma B.1.** *For any $t > 0$, the magnitude distance satisfies the non-negativity axiom.*

*Proof of Lemma B.1.* Let $X, Y \subset \mathbb{R}^D$ be finite sets, and let $Z = X \cup Y$. By [Corollary 2.4.4] (Leinster, 2013), the inclusion $X \subseteq Z$ implies $\mathrm{Mag}_Z(t) \geq \mathrm{Mag}_X(t)$, and similarly $\mathrm{Mag}_Z(t) \geq \mathrm{Mag}_Y(t)$. Therefore,

$$d_{Mag}^t(X, Y) = 2 \, \mathrm{Mag}_Z(t) - \mathrm{Mag}_X(t) - \mathrm{Mag}_Y(t) \geq 0.$$

$\square$

**Corollary B.2.** *Let $X, Y \subset \mathbb{R}^D$ be finite sets with weighting vectors $\mathbf{w}_X^t$ and let $t$ be a scaling parameter. The magnitude equivalency is an equivalence relation, satisfying*

- **Reflexivity:** $X \underset{\mathrm{Mag}(t)}{=} X,$

- **Symmetry:** *If* $X \underset{\mathrm{Mag}(t)}{=} Y$, *then* $Y \underset{\mathrm{Mag}(t)}{=} X.$

- **Transitivity:** *If* $X \underset{\mathrm{Mag}(t)}{=} Y$, *and* $Y \underset{\mathrm{Mag}(t)}{=} Z$, *then* $X \underset{\mathrm{Mag}(t)}{=} Z.$

*Proof of Corollary B.2.* Let $\mathbf{w}_X^t$, $\mathbf{w}_Y^t$ and $\mathbf{w}_Z^t$ be weighting vectors at scale $t$ for $X, Y$, and $Z$ respectively. The first two properties are trivially true by definition. If $X \underset{\mathrm{Mag}(t)}{=} Y$, and $Y \underset{\mathrm{Mag}(t)}{=} Z$, then

$$\{x \in X : \mathbf{w}_X^t(x) \neq 0\} = \{y \in Y : \mathbf{w}_Y^t(y) \neq 0\},$$
$$\text{and } \{y \in Y : \mathbf{w}_Y^t(y) \neq 0\} = \{z \in Z : \mathbf{w}_Z^t(z) \neq 0\}.$$

Thus,

$$\{x \in X : \mathbf{w}_X^t(x) \neq 0\} = \{z \in Z : \mathbf{w}_Z^t(z) \neq 0\}.$$

$\square$

**Lemma B.3.** *Let $X, Y \subset \mathbb{R}^D$ be finite sets, and let $\mathbf{w}_X^t$, and $\mathbf{w}_Y^t$ denote the weighting vectors of $X$, and $Y$ at scale $t$, respectively. Then,*

$$X \underset{\mathrm{Mag}(t)}{=} Y \quad \Longleftrightarrow \quad \begin{cases} \mathbf{w}_X^t(z) = \mathbf{w}_Y^t(z) & \text{if } z \in Q, \\ \mathbf{w}_X^t(z) = 0 & \text{if } z \in X \setminus Q, \\ \mathbf{w}_Y^t(z) = 0 & \text{if } z \in Y \setminus Q. \end{cases}$$

*where $Q = X \cap Y$.*

*Proof of Lemma B.3.* We only prove the forward direction, as the converse direction is trivial. Let $Q = \{x \in X : \mathbf{w}_X^t(x) \neq 0\} = \{y \in Y : \mathbf{w}_Y^t(y) \neq 0\}$. Build vector $\mathbf{u}_X$ and $\mathbf{u}_Y$ such that for every $q \in Q$ we have $\mathbf{u}_X(q) = \mathbf{w}_X^t(q)$ and $\mathbf{u}_Y(q) = \mathbf{w}_Y^t(q)$. We know $\mathbf{w}_X^t(x) = 0$ for every $x \in X \setminus Q$ and by definition of weighting we have $1 = \zeta_{tX}\mathbf{w}_X^t = \zeta_{tQ}\mathbf{u}_X$ for $q \in Q$. Similarly, we have $1 = \zeta_{tQ}\mathbf{u}_Y$. Therefore, both $\mathbf{u}_X$ and $\mathbf{u}_Y$ are valid weighting for $Q$. By uniqueness of weightings, we have $\mathbf{u}_X = \mathbf{u}_Y$. Note that $Q \subseteq X \cap Y$ and weights outside $Q$ vanish. $\square$

**Lemma B.4.** *Let $X, Z \subset \mathbb{R}^D$ be finite sets which $X \subseteq Z$. Then, $\mathrm{Mag}_Z(t) = \mathrm{Mag}_X(t)$ if and only if $X \underset{\mathrm{Mag}(t)}{=} Z$.*

*Proof of Lemma B.4.* By [Proposition 2.4.3] (Leinster, 2013) we have:

$$Mag(X) = \sup_{v \neq 0} \frac{\left(\sum_{x \in X} v(x)\right)^2}{v^* \zeta_{tX} v},$$

$$Mag(Z) = \sup_{u \neq 0} \frac{\left(\sum_{z \in Z} u(z)\right)^2}{u^* \zeta_{tZ} u}. \tag{7}$$

First, we prove the forward direction and assume $Mag(X) = Mag(Z)$. Define a vector $u$ on $Z$ by extending the weighting $\mathbf{w}_X^t$ of $X$:

$$\mathbf{u} = \begin{cases} \mathbf{w}_X^t(x) & \text{if } z \in X, \\ 0 & \text{if } z \in Z \setminus X. \end{cases}$$

By definition of $\mathbf{u}$ and considering the magnitude as the sum of the weighting entries, we have:

$$\begin{aligned} \mathbf{u}^* \zeta_{tZ} \mathbf{u} &= \sum_{z_1, z_2 \in Z} \mathbf{u}(z_1) e^{-td(z_1, z_2)} \mathbf{u}(z_2) \\ &= \sum_{x_1, x_2 \in X} \mathbf{w}^t(x_1) e^{-td(x_1, x_2)} \mathbf{w}^t(x_2) \\ &= \mathbf{w}_X^{t\,*} \zeta_{tX} \mathbf{w}_X^t = \mathrm{Mag}_X(t), \end{aligned} \tag{8}$$

and

$$\sum_{z \in Z} \mathbf{u}(z) = \sum_{x \in X} \mathbf{w}_X^t(x) = \mathrm{Mag}_X(t). \tag{9}$$

Therefore, $\mathbf{u}$ is the weighting on $Z$ and by substituting the equation equation (7) with equation (8), equation (9), we have:

$$\frac{\left(\sum_{z \in Z} \mathbf{u}(z)\right)^2}{\mathbf{u}^* \zeta_{tZ} \mathbf{u}} = \frac{\mathrm{Mag}_X(t)^2}{\mathrm{Mag}_X(t)} = \mathrm{Mag}_X(t). \tag{10}$$

Therefore, by considering equation (7), and equation (10) we have $\mathrm{Mag}_Z(t) = \sup_{u \neq 0} \frac{\left(\sum_{z \in Z} u(z)\right)^2}{u^* \zeta_{tz} u} \geq \mathrm{Mag}_X(t)$. Then, based on our assumption i.e., $\mathrm{Mag}_X(t) = \mathrm{Mag}_Z(t)$, the supremum is achieved at $\mathbf{u}$. By [Proposition 2.4.3] (Leinster,

2013) the weighting vector $\mathbf{u}$ must be scalar multiple of the unique weighting $\mathbf{w}_Z^t$ on $Z$, i.e. for some $c \neq 0$ we have $\mathbf{u} = c\mathbf{w}_Z^t$. Then, by definition of $\mathbf{u}$ we have $c\mathbf{w}_Z^t(z) = \mathbf{u}(z) = 0$ for $z \in Z \setminus X$ and $c\mathbf{w}_Z^t(z) = \mathbf{u}(z)$ for $z \in X$, implying $X \underset{\mathrm{Mag}(t)}{=} Z$.

While the converse direction is a direct conclusion of Lemma B.3, in the following we explain another proof. For proving the converse direction, assume $X \underset{\mathrm{Mag}(t)}{=} Z$. Then for every $x \in Z \setminus X$, we have $\mathbf{w}_Z^t(z) = 0$ where $\mathbf{w}_Z^t$ is the weighting vector of $Z$. Let $\mathbf{u}$ be a vector which $\mathbf{u}(x) = \mathbf{w}_Z^t(x)$ for every $x \in X$. Now by definition of weighting vector and $\mathbf{u}$ for any $x \in X$ we have $1 = \zeta_{tZ}\mathbf{w}_Z^t = \zeta_{tX}\mathbf{u}$ implying that $\mathbf{u}$ is a valid weighting for $X$. This completes the proof as $\mathbf{w}_Z^t(z) = 0$ for every $x \in Z \setminus X$, and

$$\mathrm{Mag}_X(t) = \sum_{x \in X} \mathbf{u}(x) = \sum_{x \in X} \mathbf{w}_Z^t(x) = \sum_{z \in Z} \mathbf{w}_Z^t(z) = \mathrm{Mag}_Z(t).$$

$\square$

**Lemma B.5.** *Magnitude distance satisfies the identity of indiscernibles, with respect to the magnitude equivalency:*

$$d_{Mag}^t(X, Y) = 0 \iff X \underset{\mathrm{Mag}(t)}{=} Y. \tag{11}$$

*Proof of Lemma B.5.* By definition of the magnitude distance equation (3), we have $d_{Mag}^t(X, Y) = 0$ if and only if $2\mathrm{Mag}_{X \cup Y}(t) = \mathrm{Mag}_X(t) + \mathrm{Mag}_Y(t)$. Let $Z = X \cup Y$. Then, by [Corollary 2.4.4] (Leinster, 2013), we know that $\mathrm{Mag}_Z(t) \geq \mathrm{Mag}_X(t)$ and $\mathrm{Mag}_Z(t) \geq \mathrm{Mag}_Y(t)$. Therefore, equality $2\mathrm{Mag}_{X \cup Y}(t) = \mathrm{Mag}_X(t) + \mathrm{Mag}_Y(t)$ holds if and only if both inequalities are tight i.e. $\mathrm{Mag}_Z(t) = \mathrm{Mag}_X(t)$ and $\mathrm{Mag}_Z(t) = \mathrm{Mag}_Y(t)$. By lemma B.4, statements $\mathrm{Mag}_Z(t) = \mathrm{Mag}_X(t)$ and $X \underset{\mathrm{Mag}(t)}{=} Z$, and also statements $\mathrm{Mag}_Z(t) = \mathrm{Mag}_Y(t)$ and $Y \underset{\mathrm{Mag}(t)}{=} Z$ are equivalent. Thus,

$$d_{Mag}^t(X, Y) = 0 \iff X \underset{\mathrm{Mag}(t)}{=} Z \text{ and } Y \underset{\mathrm{Mag}(t)}{=} Z.$$

Finally, Lemma B.3 implies that whenever $X \underset{\mathrm{Mag}(t)}{=} Y$, then we have $X \underset{\mathrm{Mag}(t)}{=} Y \underset{\mathrm{Mag}(t)}{=} Z$. The converse direction also follows from the transitivity of magnitude equivalence. $\square$

**Lemma B.6.** *Magnitude distance does not satisfy the triangle inequality in general. Moreover, the triangle inequality for the magnitude distance is equivalent to the submodularity of magnitude.*

*Proof of Theorem B.6.* Magnitude distance satisfies the triangle inequality if for every three finite sets of samples $X, Y, Z \subset \mathbb{R}^D$, if and only if we have:

$$d_{Mag}^t(X, Y) + d_{Mag}^t(Y, Z) \geq d_{Mag}^t(X, Z).$$

Substituting the definition of magnitude distance equation (3) in this inequality is equivalent to $\mathrm{Mag}_{X \cup Y}(t) + \mathrm{Mag}_{Y \cup Z}(t) - \mathrm{Mag}_Y(t) \geq \mathrm{Mag}_{X \cup Z}(t)$, which can be written as:

$$\mathrm{Mag}_{X \cup Y}(t) + \mathrm{Mag}_{Y \cup Z}(t) \geq \mathrm{Mag}_{X \cup Z}(t) + \mathrm{Mag}_Y(t). \tag{12}$$

If we take $Y = X \cap Z$ then equation (12) becomes the submodularity inequality, i.e. $\mathrm{Mag}_X(t) + \mathrm{Mag}_Z(t) \geq \mathrm{Mag}_{X \cup Z}(t) + \mathrm{Mag}_{X \cap Z}(t)$. Conversely, define $X' = X \cup Y$ and $Z' = Z \cup Y$, then the submodularity equation for $X'$ and $Z'$ will be $\mathrm{Mag}_{X'}(t) + \mathrm{Mag}_{Z'}(t) \geq \mathrm{Mag}_{X' \cup Z'}(t) + \mathrm{Mag}_{X' \cap Z'}(t)$ which is

$$\mathrm{Mag}_{X \cup Y}(t) + \mathrm{Mag}_{Z \cup Y}(t) \geq \mathrm{Mag}_{(X \cup Y) \cup (Z \cup Y)}(t) + \mathrm{Mag}_{(X \cup Y) \cap (Z \cup Y)}(t).$$

Equivalently, $\mathrm{Mag}_{X \cup Y}(t) + \mathrm{Mag}_{Z \cup Y}(t) \geq \mathrm{Mag}_{X \cup Y \cup Z}(t) + \mathrm{Mag}_{Y \cup (X \cap Z)}(t)$ which by [Lemma 2.4.12] (Leinster, 2013) we have $\mathrm{Mag}_{X \cup Y \cup Z}(t) \geq \mathrm{Mag}_{X \cup Z}(t)$ and $\mathrm{Mag}_{Y \cup (X \cap Z)}(t) \geq \mathrm{Mag}_Y(t)$. This implies

$$\mathrm{Mag}_{X \cup Y}(t) + \mathrm{Mag}_{Z \cup Y}(t) \geq \mathrm{Mag}_{X \cup Y \cup Z}(t) + \mathrm{Mag}_{Y \cup (X \cap Z)}(t)$$

which is the equation (12). Therefore, the triangle inequality for the magnitude distance is equivalent to the submodularity of magnitude. Submodularity is known not to hold for magnitude in general (Andreeva et al., 2025). Now, we show a counter-example that equation (12) does not hold in general using [Theorem 2] (Andreeva et al., 2025). Let $Y = \emptyset$ and $Z = \{0\}$. Then $\text{Mag}_Y(t) = 0$, $\text{Mag}_Z(t) = 1$, and $X \cup Y = X$. Then, $X$ is build on $S = \{e_1, \cdots, e_D\}$ which is the standard basis vectors of $\mathbb{R}^D$ as $X = \{te_1, -te_1, \cdots, te_D, -te_D\}$. For these choices, equation (12) is:

$$\text{Mag}_X(t) + 1 \geq \text{Mag}_{X \cup Z}(t),$$

which this inequality does not hold with the appropriate choice of $t$ and $D$. For instance, when $t = 5$ and $D = 500$, $\text{Mag}_{X \cup Z}(5) - \text{Mag}_X(5) \approx 7.18$. $\qquad \square$

**Proposition B.7.** *Magnitude distance satisfies the triangle inequality in $\mathbb{R}$.*

*Proof of Proposition B.7.* By Lemma B.6, the triangle inequality for the magnitude distance is equivalent to the submodularity of the magnitude. While submodularity is known not to hold for magnitude in general, it does hold in the one-dimensional case $\mathbb{R}$ [Theorem 3] (Andreeva et al., 2025). $\qquad \square$

## C. Proofs: Magnitude Distance Over Different $t$ Values

*Proposition* (2.2.6, (Leinster, 2013)). Let $X$ be a finite metric space. Then:

- For $t \gg 0$, the magnitude function of $X$ is increasing with respect to $t$.

- For $t \gg 0$, there is a unique, positive, weighting on $tX$.

- Converges to the cardinality of $X$ as $t \to \infty$.

**Theorem 5.3.** *For every finite metric sets $X$ and $Y$, the magnitude distance $d_{Mag}^t(X, Y)$:*

- *Converges to 0 as $t \to 0$.*

- *Converges to the cardinality of $X \Delta Y$ as $t \to \infty$.*

- *For $t \gg 0$, the magnitude distance $d_{Mag}^t(X, Y)$ is increasing with respect to $t$.*

*Proof of Theorem 5.3.* The first two statements follow directly from Proposition[2.2.6] (Leinster, 2013). Now we prove the third one as follows. Let $M_{tX} = \zeta_{tX} - I$ be a matrix, where $I$ is the identity matrix. Therefore, $M_{tX}$ has zero diagonal as $\exp(-t\, d(x_i, x_i)) = 1$ for all $x_i \in X$. Also, off-diagonal entries are equal to similarity matrix and for $t \gg 0$, the off-diagonal entries of $M_{tX}$ are exponentially small, i.e., $\exp(-t\, d(x_i, x_j)) = O(\exp(-td_{\min}(X)))$ where $d_{\min}(X) = \min\limits_{x_i \neq x_j} d(x_i, x_j)$.

By $t \to \infty$, matrix $\|M_{tX}\|$ converges to zero and for sufficiently large $t$, the the spectral radius of $M_{tX}$ is less than 1. Then, $\zeta_{tX}^{-1}$ can be expanded using the converging Neumann series,

$$\zeta_{tX}^{-1} = (I + M_{tX})^{-1} = \sum_{k=0}^{\infty} (-1)^k M_{tX}^k.$$

Now recalling the definition of magnitude, using Neumann series, we have

$$\begin{aligned}
\text{Mag}_X(t) = \mathbf{1}^\top \zeta_{tX}^{-1} \mathbf{1} &= \sum_{k=0}^{\infty} (-1)^k \mathbf{1}^\top M_{tX}^k \mathbf{1} \\
&= |X| - \sum_{x_i \neq x_j, x_i, x_j \in X} e^{-t\, d(x_i, x_j)} + O(e^{-2td_{\min}(X)}).
\end{aligned}$$

and the derivative is

$$\frac{d}{dt}\operatorname{Mag}_X(t) = \sum_{x_i \neq x_j, x_i, x_j \in X} d(x_i, x_j)e^{-t\,d(x_i,x_j)} + O(e^{-2td_{\min}(X)}).$$

Similarly, defining the magnitude of $Y$ and $X \cup Y$ using Neumann series, the derivative of the magnitude distance can be written as

$$\begin{aligned}
\frac{d}{dt}d^t_{\operatorname{Mag}}(X,Y) &= 2\frac{d}{dt}\operatorname{Mag}_Z(t) - \frac{d}{dt}\operatorname{Mag}_X(t) - \frac{d}{dt}\operatorname{Mag}_Y(t)\\
&= 2\sum_{z_i \neq z_j, z_i, z_j \in X \cup Y} d(z_i, z_j)e^{-t\,d(z_i,z_j)} - \sum_{x_i \neq x_j, x_i, x_j \in X} d(x_i, x_j)e^{-t\,d(x_i,x_j)}\\
&\quad - \sum_{y_i \neq y_j, y_i, y_j \in Y} d(y_i, y_j)e^{-t\,d(y_i,y_j)} + O(e^{-2td_{\min}(X \cup Y)}).
\end{aligned}$$

This can be simplified to

$$\begin{aligned}
\frac{d}{dt}d^t_{\operatorname{Mag}}(X,Y) &= 2\sum_{x_i \neq y_j, x_i \in X, y_j \in Y} d(x_i, y_j)e^{-t\,d(x_i,y_j)} + \sum_{x_i \neq x_j, x_i, x_j \in X} d(x_i, x_j)e^{-t\,d(x_i,x_j)}\\
&\quad + \sum_{y_i \neq y_j, y_i, y_j \in Y} d(y_i, y_j)e^{-t\,d(y_i,y_j)} + O(e^{-2td_{\min}(X \cup Y)}),
\end{aligned}$$

where each term is nonnegative. This indicates, for sufficiently large $t$, magnitude distance is positive, up to exponentially small error. $\qquad\square$

## D. Plots: Performance in High dimensions

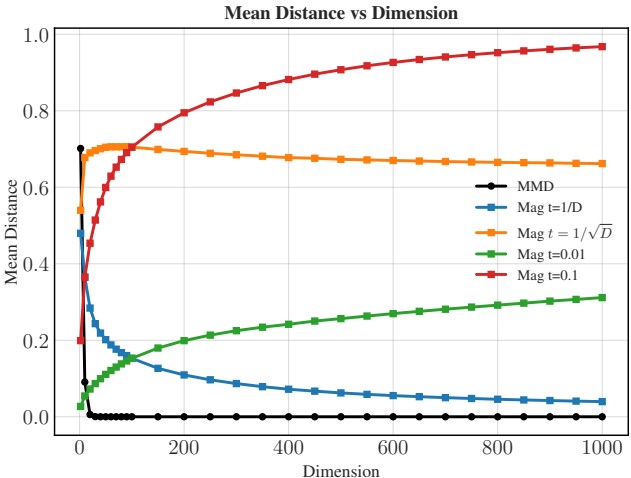

*Figure 7.* From two Gaussian distributions with identical covariance and a mean shift of $2\sqrt{D}$, we compute the empirical MMD and magnitude distance between 500 samples from each distribution over 100 independent trials. The plot shows a comparison between the mean of empirical MMD distance with Gaussian kernel bandwidth $\sigma = 1$ and normalized magnitude distance for fixed kernel scales $t \in \{0.01, 0.1\}$ and adaptive scales $t = 1/D$ and $t = 1/\sqrt{D}$. MMD with fixed bandwidth rapidly collapses toward zero as dimension increases, magnitude distance with constant $t$ grows with dimension, and magnitude scaling ($t = 1/D$) leads to gradual collapse. In contrast, scaling $t = 1/\sqrt{D}$ shows a stable behavior across dimensions.

## E. Proofs: Outlier Robustness

**Theorem 5.5.** *Let $X, Y \subset \mathbb{R}^D$ be finite sets with nonnegative weighting vectors of $X, Y$, and $X \cup Y$. Then, we have:*

$$0 \leq d^t_{\operatorname{Mag}}(X,Y) \leq 2(|X \cup Y|). \tag{5}$$

*where $|X|$ and $|Y|$ denote the number of points in $X$ and $Y$ respectively.*

*Proof of Theorem5.5.* The proof follows from [Lemma 2.4.12] (Leinster, 2013), which proves $\mathrm{Mag}_{X \cup Y}(t) < |X \cup Y|$, and from [Proposition 2.2.6] (Leinster, 2013), which guarantees $\mathrm{Mag}_X(t), \mathrm{Mag}_Y(t) \geq 0$, under the assumption of nonnegative weighting vectors for $X, Y$, and $X \cup Y$. Note that this assumption is satisfied for sufficiently large $t$, i.e., $t \gg 0$, by [Proposition 2.2.6] (Leinster, 2013). $\qquad\square$

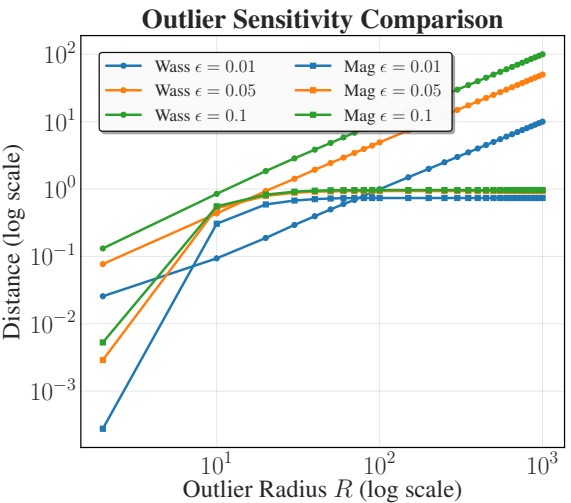

*Figure 8.* Outlier robustness under Huber contamination. We compare empirical Wasserstein distance and magnitude distance with $t = 0.001$ between 500 samples drawn from a contaminated distribution $Q = (1 - \epsilon)P + \epsilon R$ and 500 from the clean distribution $P$. The plot shows three contamination levels, $\epsilon \in \{0.01, 0.05, 0.1\}$. As the outlier radius increases, the Wasserstein distance grows, while normalized magnitude distance converges to a to a finite limit bounded by 1.

# F. Experiments: MagGN on Complex Datasets

Distance-based generative models suffer from drawbacks such as mode collapse due to their rich semantic variability. In this section, we examine how training adjustments allow magnitude distance-based objectives to provide useful learning signals beyond simple benchmarks.

### F.1. CIFAR-10

As a moderately complex natural image benchmark, we first evaluate MagGN on a subset of the CIFAR-10 dataset consisting of cat images. MagGn, trained directly in raw-pixel space, is able to capture low-frequency texture and color distributions. However, we are unable to tune $t$ sufficiently to capture object-level structures or generate sharp images without mode collapse. While generating sharp images with MagGN is challenging due to the limitations of purely metric-based losses, we leverage MagGN's training efficiency and employ a two-phase transfer learning strategy to initialize WGAN-GP with pretrained MagGN weights.

In Phase 1 (epochs 1-100), we initialize the generator with weights from a pretrained MagGN model and freeze these weights, training only the WGAN-GP critic to establish a stable Wasserstein distance estimate without disrupting the generator's learned features. In Phase 2 (epochs 101-600), we unfreeze the generator and train both networks in the WGAN-GP setting to fine-tune the pretrained weights. As reported in Table 2 and Figure 9, the pretrained model not only achieves visually comparable image quality and a higher Inception Score (IS), while also reduces the total training time compared to training WGAN-GP from random initialization. In both Figure 9 and Table 2, pretraining stage for the generator has MagGN (normalized) with $t = \{0.5, 1.5, 3.0, 5.0\}$ evaluated at $t$-steps $\{1, 201, 501, 701\}$ trained for 1000 epochs.

### F.2. Celeb-A

To enhance performance, we incorporate MagGN into feature spaces to capture effective attributes employing a pre-trained VGG-16 network () truncated at the third ReLU layer (ReLU3_3) to extract facial features. However, training directly in these high-dimensional feature spaces increases the dimensionality and the computational cost of magnitude distance calculations.

*Table 2.* **Training time and Inception Score comparison on CIFAR-10 (Cats)**

| Method | Training Time (s) | IS |
|---|---|---|
| WGAN-GP (1600 epochs) | **9098** | $2.886 \pm 0.149$ |
| MagGN pretraining (1000 epochs) | 2596 | – |
| WGAN-GP fine-tuning (600 epochs) | 2414 | – |
| **Total (MagGN + WGAN-GP)** | **5010** | $3.064 \pm 0.090$ |

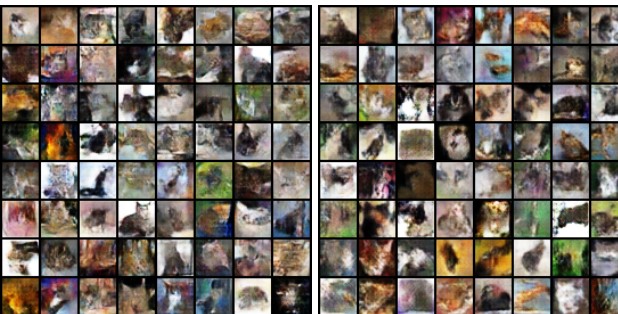

*(a)* WGAN-GP trained for 1600 epochs

*(b)* WGAN-GP pretrained with MagGN weightings, trained for 600 epochs

*Figure 9.* CIFAR-10 samples generated using WGAN-GP. (a) WGAN-GP trained from random initialization for 1600 generator epochs. (b) WGAN-GP initialized with a pretrained MagGN generator and fine-tuned for 600 generator epochs. Pretraining the generator using magnitude distance leads to improved visual quality while substantially reducing total training time.

Also, it has a high risk of overfitting to spatial structures in the features, such as low-level artifacts or dataset-specific noise. To address these issues, we apply global average pooling with an $8 \times 8$ kernel to the extracted features, which creates a suitable balance that preserves spatial structure while removing some noise, intuitively similar to how dropout improves generalization. Finally, for further refinement and fix poor spatial alignment or unnatural pixel configurations, we introduce a hybrid loss that blends feature- and pixel-level magnitudes, and at epoch $e$, the loss is

$$l_e = \sum_{i:e_i \leq e} \hat{d}_{Mag}^{t_i}(\phi_{8 \times 8}(\mathcal{D}_r), \phi_{8 \times 8}(\mathcal{D}_g)) + \gamma \hat{d}_{Mag}^{t_i}(\mathcal{D}_r, \mathcal{D}_g)$$

where $\phi_{8 \times 8}$ denotes the composition of the pre-trained VGG-16 feature extractor (truncated at ReLU3_3) followed by global average pooling with an $8 \times 8$ kernel. Table 3 reports that using the hybrid loss significantly improves Frechet Inception Distance (FID), reducing it from 32.34 (feature-only training) to 24.32 when $\gamma = 0.2$. Both models reported in the table are MagGN (normalized) trained for 400 epochs, with $t = \{0.0003, 0.009, 0.06, 0.3\}$ evaluated at $t$-steps $\{1, 201, 501, 701\}$. One model was trained using a hybrid loss with $\gamma = 0.2$.

*Table 3.* **FID scores on CelebA using MagGN**

| Method | $t$ values | Hybrid coeff. $\gamma$ | FID |
|---|---|---|---|
| MagGN (feature-only) | $\{0.0003,\ 0.009,\ 0.06,\ 0.3\}$ | 0 | 32.34 |
| MagGN (hybrid) | $\{0.0003,\ 0.009,\ 0.06,\ 0.3\}$ | 0.2 | 24.32 |

