# OpenReview forum: "Magnitude Distance: A Geometric Measure of Dataset Similarity"
_ICML.cc/2026/Conference — ICML 2026 regular_

### Official Review · Reviewer_mfZv · 2026-02-28

**Soundness:** 3
**Presentation:** 3
**Significance:** 2
**Originality:** 3
**Overall Recommendation:** 4
**Confidence:** 3

**Summary:**

This paper proposes a novel dataset distance metric, Magnitude Distance. It demonstrates that, in high dimensions, Magnitude Distance can dynamically adjust its sensitivity to global structure or local details while maintaining strong discriminative power.

**Compliance With Llm Reviewing Policy:**

Affirmed.

**Key Questions For Authors:**

Due to its high complexity, more application scenarios for Magnitude Distance need to be identified to demonstrate its potential impact.

**Limitations:**

yes, the authors point out that the high computational complexity may lead to potential limitations in scalability.

**Strengths And Weaknesses:**

Strengths:
1. To my knowledge, introducing the concept of Metric Magnitude to measure dataset similarity is a novel idea.
2. Theoretically and experimentally, it has been proven that this distance metric can effectively alleviate the curse of dimensionality in high-dimensional spaces and is superior to the classical Wasserstein distance.

Weaknesses:
1. The core computation involves inverting the N × N similarity kernel matrix, which has a time complexity of $O(N^3)$.
2. In MagGN experiments, it encountered optimization difficulties with the complex CIFAR-10 dataset, and its generalization ability seems insufficient.

---

> ### Author Rebuttal · Authors · 2026-03-31
>
> We would like to thank the reviewer for taking the time to provide this feedback.
>
> **Computational Scalability:
> MagGN is 10–12x faster than WGAN-GP on MNIST (Table 1).** While matrix inversion is $O(n^3)$, on modern GPUs, for standard mini-batches (n=512), this cost is negligible compared to the forward/backward passes of deep networks. MagGN is a single-step "push-forward" model which adds to its efficiency.
>
>
> **Empirical Validations:
> MagGN is introduced as a proof-of-concept; it can be an efficient tool for enhancing traditional training pipelines.** The paper's main goal is to introduce a novel distance metric with strong theoretical properties. MagGN is introduced as a promising direction for future research, rather than a direct competitor to the state of the art. However, on MNIST, MagGN achieves comparable quality 10–12x faster than WGAN-GP. On more complex datasets, using MagGN as a pre-training stage cuts total training time by approximately 2x while simultaneously improving final performance.
>
>
>
> **Applications (Key Question):
> Magnitude Distance is a fundamental quantity that can be applied to many ML applications that require divergences or similar measures.** While MagGN is a proof-of-concept, Magnitude captures "intrinsic cardinality" of point sets. Magnitude distance can possibly be applied to many ML scenarios where point set divergences are used.
>
> These include **Accelerating Diffusion Models:** Instead of KL divergence or similar measures, Magnitude Distance can be used as a more "geometry aware" objective. As observed in our experiments, MagGN can efficiently provide latent space initialization. This allows Diffusion models to converge in fewer steps and reduces the cost of training large-scale generative systems.
>
> **Fine-Tuning LLMs:** use of Magnitude Distance instead of divergences as a geometric regularizer which preserves the underlying manifold structure of the base model.
>
> **Privacy-Preserving ML:** Magnitude distance, being sensitive to the "distinctness" of points, can possibly be used to guide privacy-preserving mechanisms.

---

> > ### Author Rebuttal · Reviewer_mfZv · 2026-04-01
> >
> > Based on the author's feedback, most of my concerns have been addressed. Nevertheless, I believe the current rating is justified.

---

> > > ### Author Response · Authors · 2026-04-08
> > >
> > > We appreciate the time taken to review our work and for the acknowledgment that our responses have fully addressed the concerns raised.

---

### Official Review · Reviewer_nSiB · 2026-03-07

**Soundness:** 2
**Presentation:** 3
**Significance:** 2
**Originality:** 3
**Overall Recommendation:** 3
**Confidence:** 4

**Summary:**

This paper introduces magnitude distance, a new measure of dissimilarity between finite datasets based on the concept of magnitude of metric spaces. The authors extend the definition of magnitude to handle duplicate points (Theorem 4.1) and propose two variants: the raw magnitude distance dMagtd  and a normalized version. They analyze its theoretical properties, including non-negativity, identity of indiscernibles under a magnitude equivalence relation, failure of the triangle inequality in general, and behavior as the scale parameter t and to the size of the symmetric difference as t→∞. The paper also studies the robustness to outliers (Theorem 5.5) and the relationship with MMD in high dimensions. As a proof-of-concept application, the authors propose MagGN, a push-forward generative model trained by minimizing multi-scale magnitude distances, and demonstrate it on MNIST and CIFAR-10.

**Compliance With Llm Reviewing Policy:**

Affirmed.

**Key Questions For Authors:**

see weaknesses

**Limitations:**

--

**Strengths And Weaknesses:**

**Strengths:**

The paper introduces a conceptually new distance between datasets rooted in the mathematical theory of magnitude, which has been studied in geometry and topology but is underexplored in machine learning.

The authors provide thorough proofs of fundamental properties (metric axioms, scale behavior, outlier bounds) and extend magnitude to handle duplicate points.

The discussion linking magnitude distance to MMD (via kernel matrix spectra) and Wasserstein distance (via robustness) helps position the work within the broader context of distributional distances.

**Weaknesses:**

Limited Empirical Validation: Experiments are confined to small-scale datasets (MNIST, CIFAR-10) and simple architectures. The claimed advantages in training speed are not accompanied by comparisons with other fast generative models (e.g., GANs with spectral normalization, StyleGAN) or modern diffusion models. The CIFAR-10 results require a two-stage training (MagGN pretraining + WGAN-GP fine-tuning), which complicates the interpretation of the speedup.

Computational Scalability: The magnitude distance requires inverting an n×n similarity matrix (where n is the dataset size or batch size), which has O(n3) complexity. This severely limits its applicability to large datasets or even moderate-sized batches, unless approximations are used (which are not explored). The paper does not discuss runtime for computing the distance itself, only overall training time.

Lack of Comparison with Other Distances: The generative modeling experiments compare only with WGAN and WGAN-GP. It would be informative to see how MagGN performs against other distributional distances (e.g., MMD-GAN, Sinkhorn GAN) or against metrics like FID directly optimized via feature matching.

Failure of Triangle Inequality: While acknowledged, the lack of triangle inequality limits the use of magnitude distance in applications that require a proper metric (e.g., clustering with centroid-based methods, hierarchical clustering). The paper does not discuss the practical implications of this failure.

Outlier Robustness Assumptions: Theorem 5.5 relies on non-negative weightings, which are guaranteed only for sufficiently large t. For smaller t, the bound may not hold, and the robustness claim is less certain.

---

> ### Author Rebuttal · Authors · 2026-03-31
>
> We would like to appreciate the reviewer for taking the time to provide this detailed feedback and recognition of our work as a conceptually new approach.
>
> **Computational Scalability:
> MagGN is 10–12x faster than WGAN-GP on MNIST (Table 1).** While matrix inversion is $O(n^3)$, on modern GPUs, for standard mini-batches (n=512), this cost is negligible compared to the forward/backward passes of deep networks. This is empirically validated by our results. MagGN is a single-step "push-forward" model which helps its efficiency.
>
>
> **Limited Empirical Validation:
> MagGN is introduced as a proof-of-concept; it is an efficient tool for enhancing traditional training pipelines.** The paper's main goal is to introduce a novel distance metric with strong theoretical properties. MagGN is introduced proof of concept that this measure is useful, rather than a direct competitor to highly engineered SOTA architectures. On more complex datasets, we show that using MagGN as a pre-training stage cuts total training time by approximately 2x while simultaneously improving final performance.
>
>
> **Triangle Inequality:
> The absence of the triangle inequality is a necessary compromise required to capture distances between complex datasets.** MMD, KL divergence, and Bregman divergences in general do not satisfy the triangle inequality, but are effective optimization objectives.
>
>
> **Outlier Robustness Assumptions:**
> We agree that the non-negativity of the weighting is only guaranteed for large $t$. However, by Theorem 5.3, at small values of $t$, the entire distance converges to zero, and consequently, the sensitivity to outliers also converges to zero in this regime. We will amend the section to highlight this behaviour.
>
>
> **Comparison with Other Distances:
> We provide direct comparisons of Magnitude Distance against other distributional metrics, but MagGN is conceptually distinct.** Figures 2, 3, 5, 7, and 8 demonstrate comparisons of Magnitude Distance with other metrics. However, beyond the distance itself, for MagGN, while we use standard baselines, WGAN, and WGAN-GP, MagGN is conceptually distinct. Unlike WGAN, MMD-GAN, and Style-GAN, which are adversarial networks prone to mode collapse, MagGN is non-adversarial, which makes the approach inherently more efficient. Opposed to kernel-based methods (like MMD) that are highly sensitive to a single bandwidth, MagGN employs multi-scale magnitude distances. By increasing the scale to capture finer details, our method remains robust to hyperparameter choices and avoids the extensive "manual tuning" typically required by traditional kernel-based methods.

---

> > ### Author Rebuttal · Reviewer_nSiB · 2026-04-08
> >
> > Some of my concerns (e.g. triangle inequality) remain.

---

### Official Review · Reviewer_c4MU · 2026-03-09

**Soundness:** 3
**Presentation:** 3
**Significance:** 2
**Originality:** 2
**Overall Recommendation:** 5
**Confidence:** 4

**Summary:**

This paper suggests and investigates a notion of magnitude distance as a measure of discrepancy between data sets, drawing on the mathematical notion of the magnitude of a metric space. The paper presents various theoretical properties of magnitude distance and illustrates its use in generative modelling.

**Compliance With Llm Reviewing Policy:**

Affirmed.

**Final Justification:**

The authors rebuttal fully addressed my comments and questions so I have raised my overall score 4 to 5.

**Key Questions For Authors:**

I am just wondering if, in the experiments presented, the authors can offer any more intuitive explanation of why the proposed distance performs well.

**Limitations:**

Societal impacts are briefly touched upon, this seem appropriate concerning the nature of the work.

I didn't find any explicit mention of weaknesses or disadvantages, did the authors find any experimental settings in which their approach did not, in some sense, work "well"?

**Strengths And Weaknesses:**

As far as I can tell the paper is technically sound. I wondered if the authors would be able to offer some intuition (especially to an ML/AI crowd) about what magnitude distance captures. The discussion in relation to MMD is informative, but didn't leave me feeling like I had a clear interpretation of what magnitude distance measures.

The mathematical presentation is clear and easy to read. I did some "spot checks" on the proofs in places, and didn't see any problems. The layout and organisation of the paper is mostly logical, I did wonder if fig4 and fig5 could be made smaller and still easily convey the same information.

The main strength of the paper seems to be suggesting and studying the notion of magnitude distance. This could potentially be significant if magnitude distance can be demonstrated to be especially useful in some application domain. I found myself wondering what a "killer" application for magnitude distance might be. Whilst the generative modelling experiments (including those in the appendix) do demonstrate some utility in terms of numerical performance metrics, I would have preferred to see more investigation of _why_ these performance improvements might be possible. As such it feels a bit like the potential impact of magnitude distance remains to be demonstrated.

As far as I know, this concept of distance is a new tool for ML/AI, etc., and in this sense the paper does show originality, both in terms of studying magnitude distance and in signposting ML/AI researchers to the underlying mathematics.

---

> ### Author Rebuttal · Authors · 2026-03-31
>
> We would like to thank the reviewer for taking the time to provide this feedback. We will optimize the figure sizes and refine the introduction to clarify the underlying geometric intuitions as suggested.
>
> **Intuition behind Magnitude Distance:**
>
> Magnitude measures the "intrinsic cardinality" of a metric space at a specific scale. It effectively quantifies the number of "distinct" points in the space at a given scale, where points clustered together are counted for less than their base cardinality. Magnitude distance inherits and makes use of this property, so that if point $x$ in set $A$ is very close to point $y$ in set $B$, then in the union they are counted as “almost the same point” and thus add to the similarity between the sets. Consequently, MagGN also optimizes the generator to recreate the geometric density of the target manifold through a multi-scale training process, capturing increasingly structural details as the scaling parameter is adjusted.
>
>
> **Application:**
> We agree that the applications of Magnitude Distance require further study. Given that it defines a fundamental quantity, we expect it to be a useful tool in a variety of ML domains. While MagGN is introduced as a proof-of-concept, it demonstrates a significant 10–12x speedup over WGAN-GP on MNIST and a 2x total improvement when utilized for pre-training. These results are not intended to position MagGN as a standalone generative approach; instead, they serve as evidence that magnitude distance can function as a robust, theory-supported component for regularizing more complex, high-resolution architectures.
>
>
> **Limitations Section:**
> A separate limitations section was not required at submission. We have addressed the theoretical limitations of Magnitude Distance, including the scaling parameter $t$, the triangle inequality, and computational complexity, directly within their respective sections. Regarding the practical boundaries of MagGN, we have included a detailed discussion in the Appendix concerning its performance on high-resolution datasets like Celeb-A.

---

> > ### Author Rebuttal · Reviewer_c4MU · 2026-04-02
> >
> > Thanks to the authors for their response which addresses my question, I will raise my score accordingly.

---

> > > ### Author Response · Authors · 2026-04-08
> > >
> > > We sincerely appreciate the time taken to review our work and and adjusting our score. We are glad that our responses addressed your questions.

---

### Official Review · Reviewer_xF2j · 2026-03-22

**Soundness:** 3
**Presentation:** 3
**Significance:** 2
**Originality:** 3
**Overall Recommendation:** 3
**Confidence:** 3

**Summary:**

This work proposes a novel distance metric for measuring dataset similarity by leveraging the magnitude in metric spaces. The authors provide theoretical support for its axiomic properties. Through outlier analysis and comparisons with MMD and Wasserstein distance, the proposed metric demonstrated robustness in a noisy dataset.  Finally, the authors introduce MagGN and validate its effectiveness on the MNIST dataset.

**Compliance With Llm Reviewing Policy:**

Affirmed.

**Key Questions For Authors:**

Are there any restrictions on the choice of the similarity kernel? And how might different kernels affect the proposed metric's performance?

**Limitations:**

The limitations section is missing.

**Strengths And Weaknesses:**

Strengths:
- Clear statement and theoretical proofs.
- Provide and prove axioms of the proposed metric.
- Robust analysis and comparisons for outliers' influence on metrics' performance.

Weaknesses:
-  The performance of the proposed metric $d^t_{MAG}$ is highly determined by the scaling factor $t$, but there is a lack of optimal $t$ value tuning guidance. In complex data, multiple $t$ values are required, which makes it unclear whether this is applicable in practice.
-  Push-Forward Generative Modelling seems to replace Wasserstein distance with $d^t_{MAG}$ in GAN's loss function. However, according to Figure 6, the advantage over WGAN-GP is unclear.
- Performance comparisons by comparing the quality of generated images are missing among MagGN, WGAN, and WGAN-GP, so it is unclear whether $d^t_{MAG}$ can improve the generation performance compared with the existing metrics.
- The experiment is only conducted on MNIST, which is not sufficient.

---

> ### Author Rebuttal · Authors · 2026-03-31
>
> We would like to thank the reviewer for taking the time to provide this feedback. We are glad you found our proofs and analysis clear and robust. Below, we address your comments.
>
> **Performance comparisons with WGAN and WGAN-GP (Weakness 2 and 3):
> MagGN reaches comparable quality to WGAN-GP 10–12x faster, as shown in Table 1.** We will amend the phrasing and directly refer to Figure 6 which shows comparable final visual quality to WGAN-GP and significantly better than WGAN while Table 1 shows that MagGN reaches this performance faster than both WGAN and WGAN-GP.  Further comparisons are present in the Appendix: Table 3 reports Inception Score (IS) for CIFAR-10, and Section F.2 shows improved Fréchet Inception Distance (FID) for Celeb-A.
>
>
> **Limited experiments (Weakness 4):
> We provided CIFAR-10 and Celeb-A results in the Appendix to maintain the main body's focus on our theoretical contributions.** While the main body focuses on MNIST for clarity of the proof-of-concept and presents MagGN as a promising direction for future research, we would like to highlight the provided results on CIFAR-10 (with two-stage training to mitigate mode collapse) and CelebA (incorporating into feature spaces) in the Appendix. We will add more explicit pointers to these results in the main text to ensure they are not overlooked.
>
>
> **Tuning of parameter t (weakness 1st):
> The Magnitude Distance is robust to the choice of $t$, and using multiple scales further simplifies the tuning process.** Section 7.2, 'Tuning of Scaling Parameter $t$', provides both empirical and theoretical guidance for selecting a reasonable scaling, we will reference it earlier in the paper. To address your concern about multi-scales, Magnitude Distance is stable across multiple values of $t$, as seen in different choices of $t$ values in the experiments that yield close results. We will amend the phrasing to emphasize more that multiple scaling parameters can facilitate tuning, and the performance remains robust even if the magnitude is not perfectly tuned.
>
>
> **Kernel Choice (Key Question):**
> Magnitude is defined as the cardinality of an enriched category, which imposes the requirements of isomorphism invariance and multiplicativity. This is shown in Leinster et al. (2021) “Entropy and diversity The axiomatic approach”. When defining the magnitude of a finite V-category, and in particular of a finite metric space, the multiplicativity property essentially forces an exponential similarity matrix, specifically $e^{-d(ai,aj)}$. This kernel is thus compatible with the mathematical development of magnitude and has known useful properties (e.g. magnitude is positive for points in Eucledian spaces). We will add an explanation of this design choice and its theoretical grounding in Section 3 (or the appendix) to explain this theoretical basis.
>
>
> **Limitations Section:
> A separate limitations section was not required at submission in the call for papers.** Various design considerations, such as tuning the scaling parameter $t$, the triangle inequality, and computational complexity, are discussed directly alongside the theoretical sections where they arise. We have also provided the required "Impact Statement."

---

> > ### Author Rebuttal · Reviewer_xF2j · 2026-04-03
> >
> > Thank you for the explanation. My concerns remain: even though efficient computation is an advantage, it is difficult to clearly distinguish improvements in image quality in Figure 6. It would be better represented in a quantitative way. Additionally, the current evaluation is limited to the vision domain.

---

> > > ### Author Response · Authors · 2026-04-08
> > >
> > > We appreciate the time taken to review our work and for the follow-up. We would like to clarify our results:
> > > The distinguishable visual improvement is only considered comparing to WGAN which struggled to generate legible digits. Regarding WGAN-GP, we agree there is no “clear distinguishable visual improvement”, however our key contribution is the 10-12 times increase in the computational efficiency demonstrated in Table 1.
> > > Quantitative validation showing the improvement using two metrics, FID and IS, is provided in the Appendix, Tables 2 and 3.

---

### Decision · Program_Chairs · 2026-04-30

**Decision:**

Accept (regular)

**Comment:**

This paper introduces Magnitude Distance, a new dissimilarity measure between finite datasets. The authors provide a theoretical framework, proving properties such as scale-dependent behavior, robustness to outliers, and its relationship to established metrics like MMD and Wasserstein distance. The authors train a generative model as a proof of concept, achieving comparable visual results to other methods while being faster.

I have read through all of the reviews, rebuttals, and discussions. The reviews on the paper are somewhat mixed. The main remaining concerns seemed to be:
* Empirical scope. One reviewer felt that the authors should include other non-vision applications. I do not agree with this assessment and believe that the empirical demonstrations were sufficient for the scope of the paper. None of the other reviewers seemed to share this concern.
* Lack of triangle inequality. I agree with the authors point here that many other divergences are used in ML that also lack the triangle inequality. Thus this is not sufficient for rejection in my opinion.

Overall, I believe this is a solid contribution to the ML community.